# Clinical characteristics and optical coherence tomography findings in epiretinal membrane, macular pseudohole, epiretinal membrane-foveoschisis, and lamellar macular hole

Noriko Kubota[1]*, Kazunori Miyata[2], Yosai Mori[2], Yuji Nakano[1], Hitoshi Goto[3], Fumiki Okamoto[1]

1 Department of Ophthalmology, Nippon Medical School, Tokyo, Japan, 2 Miyata Eye Hospital, Miyazaki, Japan, 3 Department of Ophthalmology, Nippon Medical School Tama Nagayama, Tokyo, Japan

* oishinoriko@nms.ac.jp

## Abstract

### Purpose

To evaluate the optical coherence tomography (OCT) findings of epiretinal membrane (ERM) and its three associated diseases: macular pseudohole (MPH), ERM-foveoschisis (ERM-FS), and lamellar macular hole (LMH).

### Methods

We retrospectively reviewed all eyes that underwent vitrectomy with a follow-up of at least 6 months. All eyes were classified into four groups, ERM, MPH, ERM-FS, and LMH based on spectral-domain (SD) OCT findings. Factors analyzed included preoperative and postoperative best-corrected visual acuity (BCVA), presence of inner and outer retinal cysts, epiretinal proliferation (EP), and ellipsoid zone (EZ) disruption, central fovea thickness (CFT), central retina thickness (CRT), and macular volume (MV).

### Results

After enrolling 720 eyes of 664 patients, eyes were classified into four groups: ERM (592 eyes), MPH (76 eyes), ERM-FS (63 eyes), and LMH (42 eyes). BCVA significantly improved in all groups. Although preoperative BCVA was not significantly different among the four groups, postoperative BCVA was significantly worse in LMH versus ERM ($p < 0.001$). Inner and outer retinal cysts were significantly more prevalent in ERM-FS versus ERM and the other three groups, respectively. EP was significantly more frequently observed in LMH versus the other three groups ($p < 0.001$). CFT and CRT were significantly higher in ERM versus the other three groups, and MV was significantly larger in ERM than in MPH and LMH ($p < 0.05$).

**Data availability statement:** All relevant data are within the manuscript and its Supporting Information files.

**Funding:** The author(s) received no specific funding for this work.

**Competing interests:** The authors have declared that no competing interests exist.

## Conclusion

ERM had a higher CFT and CRT, and a larger MV. The postoperative BCVA was worse in LMH versus ERM, while LMH had a higher frequency of EP.

## Introduction

Recent advances in optical coherence tomography (OCT) over the past decades have greatly enhanced the visualization of the macular microstructure and help to provide more accurate diagnoses of various macular disorders. Based on OCT findings presented by an international panel of vitreoretinal experts in 2020, Hubschman et al. were able to reach a consensus regarding the definitions of macular pseudo-hole (MPH), epiretinal membrane-foveoschisis (ERM-FS), and lamellar macular hole (LMH) [1]. The creation of this uniform definition and terminology has made it possible to conduct more precise research on MPH, ERM-FS, and LMH. Several studies have recently evaluated these clinical entities and reported on the clinical features and postoperative changes found in these eyes [2–9].

Despite the established diagnostic criteria using OCT, our clinical understanding of these associated diseases remains limited. While vitrectomy, epiretinal membrane (ERM) removal, and inner limiting membrane (ILM) removal are common surgical procedures for all four disorders, each disorder is presumed to exhibit distinct clinical characteristics and varying outcomes. In addition to the clinical characteristics, several macular structures detected via OCT that satisfy more than two of the diagnostic criteria are observed in the same eye. Furthermore, identifying which macular structures can affect the visual function would be beneficial in predicting the postoperative outcomes.

The purpose of the present study was to investigate and compare the OCT findings of patients with ERM, MPH, ERM-FS, and LMH, with the goal of identifying the macular structures that influence the visual acuity and postoperative outcomes.

## Methods

This retrospective study was reviewed and approved by the Ethics Committee of Nippon Medical School (ID: M-2023–161) in accordance with the ethical review processes in our institution [10]. Written consent was provided by patients for their information to be stored in the hospital's database and to be used for research purposes. The data for research purposes were accessed between 1/3/2024 and 31/8/2024. This study was performed according to the tenets of the Declaration of Helsinki. This study enrolled 664 patients with a diagnosis of idiopathic ERM and associated diseases and who underwent vitrectomy between April 2020 and December 2023. All of the enrolled patients had to have been followed up for at least 6 months postoperatively.

Patients were classified into four groups (ERM, MPH, ERM-FS, and LMH) based on the results of the spectral-domain OCT (Spectralis version 1.8.6.0, Heidelberg Engineering GmbH, Heidelberg, Germany). In all cases, both horizontal and vertical

B-scan images were obtained. ERM was defined as the presence of hyperreflective proliferations on the surface of the ILM. The associated diseases, MPH, ERM-FS, and LMH, were classified based on the updated criteria that were established by an international panel of vitreoretinal experts in 2020 [1]. Eyes satisfying more than two of the diagnostic criteria for associated diseases in different OCT B-scan images were classified as overlapping cases and included in the analysis of each corresponding group. Exclusion criteria included patients with a previous history of vitreoretinal surgery and ophthalmic disorders, with the exception for mild cataract. Eyes with secondary ERM attributable to retinal vascular disease, uveitis and trauma were also excluded from the study.

Patient evaluations consisted of measurements of best-corrected visual acuity (BCVA) before surgery and at 6 months after surgery. BCVA was measured utilizing the Landolt Chart and expressed as a logarithm of the minimum angle of resolution (logMAR). The spherical equivalent and axial length were measured by an autorefractometer (RC-5000, Tomey Corporation, Nagoya, Japan) and ultrasonography (AL-4000, Tomey Corporation, Nagoya, Japan), respectively.

SD-OCT b-scans were used to assess the presence of the inner and outer retinal cyst, epiretinal proliferation (EP), homogeneous, isoreflective epiretinal material over the ILM, and the ellipsoid zone (EZ) disruption of the EZ line. The inner and outer retinal cysts were characterized by their hyporeflective, small, round, or elliptical structures that were located in the inner nuclear layer (INL) and outer nuclear layer (ONL), respectively. In addition, we also measured the central fovea thickness (CFT) and central retina thickness (CRT), the mean thickness within a 1 mm diameter of the central macular area and macular volume (MV), the volume within a 6 mm diameter of the central macular area, were measured with the automated function of the spectral-domain OCT.

The 25-gauge pars plana vitrectomy was performed by two vitreoretinal specialists (F.O., Y.M.) under local anesthesia in all cases. When a clinically significant cataract was observed, it was simultaneously operated on. After inducing posterior vitreous detachment and performing core vitrectomy, we injected 0.1–0.2 mL of 0.025% brilliant blue G solution gently over the macula and then washed it out with irrigation solution. After the ERM was peeled, 0.1 mL of brilliant blue G solution was applied to the macular area. Subsequently, we then completely peeled the remaining ILM within an area approximately equal to the vertical extent of the optic disc. In cases with EP, ILM was peeled toward the fovea following ERM removal, and then inverted on the fovea. The EP was embedded into the fovea defect with ILM.

The mean scores and standard deviations were calculated for age, axial length, spherical equivalent, BCVA, CFT, CRT, and MV. We used a Tukey-Kramer test to examine the difference in the values among the groups. A Wilcoxon's signed-rank test was used to analyze the differences between the pre- and postoperative BCVA. Spearman's correlation coefficient test was used to examine the correlation between the pre- and postoperative BCVA, as well as the correlation between the pre- or postoperative BCVA and the age, axial length, spherical equivalent, and OCT findings. The analysis of the categorical data in cross tables was performed using Chi-square tests. The Mann-Whitney U test was used to examine the relationships between the presence of the retinal cyst, EP, EZ disruption and the pre- and postoperative BCVA. Multiple regression analysis was used to examine the factors that significantly influenced the postoperative BCVA. All tests of association were considered statistically significant if $p < 0.05$. The analyses were carried out using SPSS Statistics (version 29.0, IBM).

## Results

This study enrolled 720 eyes of 664 patients (340 males and 324 females), with 639 eyes undergoing both cataract surgery and pars plana vitrectomy. There was no significant difference in the rate of cataract surgery among the groups ($p = 0.41$). Table 1 shows the clinical characteristics of patients with four diseases. No significant differences were noted in the age or gender ratio among the groups. Compared to the ERM, the mean axial length was significantly longer in the MPH, ERM-FS, and LMH (Fig 1A). The spherical equivalent was significantly smaller in the MPH, ERM-FS, and LMH compared to the ERM (Fig 1B).

Fig 2 shows that the changes that were observed in the pre- and postoperative BCVA across the four groups. The baseline BCVA did not significantly differ among the four groups. Improvement in the postoperative BCVA was observed

**Table 1. Clinical characteristics of epiretinal membrane (ERM), macular pseudohole (MPH), ERM-foveoschisis (ERM-FS) and lamellar macular hole (LMH).**

| | ERM | MPH | ERM-FS | LMH |
|---|---|---|---|---|
| Number of eyes | 592 | 76 | 63 | 42 |
| Age (years) | 71.7±8.2 | 71.1±7.8 | 69.6±8.8 | 71.3±8.2 |
| Gender (Men/Women) | 288/ 258 | 36/ 39 | 28/ 35 | 21/ 21 |
| Axial length (mm) | 24.0±1.4 | 24.8±2.0* | 24.8±2.0* | 24.9±1.9* |
| Spherical equivalent (diopter) | −0.94±3.0 | −2.41±4.9** | −2.32±4.9* | −2.59±4.6* |
| Preoperative BCVA | 0.290±0.26 | 0.293±0.25 | 0.294±0.30 | 0.356±0.33 |
| Postoperative BCVA | 0.028±0.19 | 0.058±0.17 | 0.067±0.24 | 0.150±0.26*** |
| Tamponade (BSS/air) | 537/ 55 | 51/ 25 | 28/ 35 | 12/ 30 |

Values are presented as the mean±standard deviation.

BCVA=best-corrected visual acuity, BSS=balanced salt solution.

Tukey-Kramer test, *p<0.05, **p<0.01, ***p<0.001.

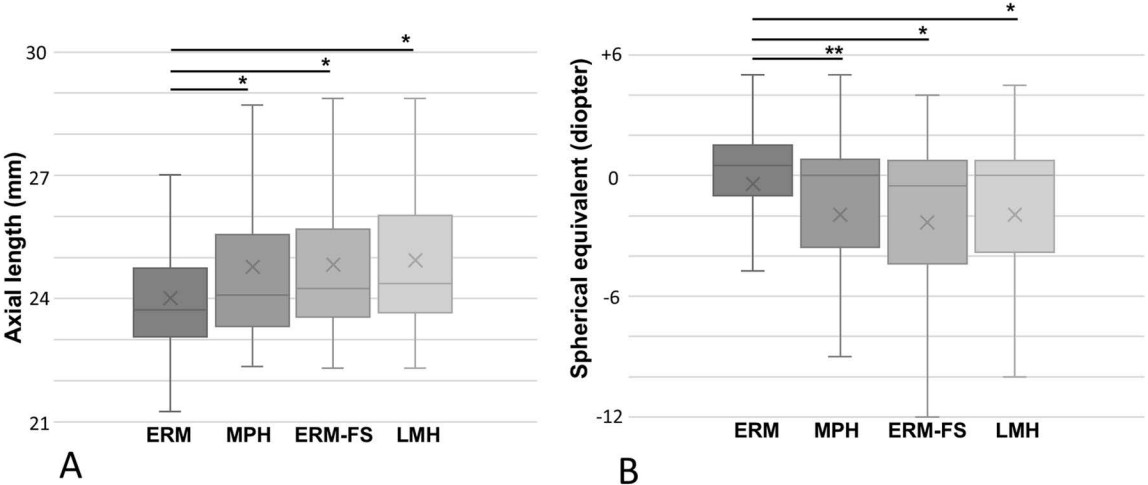

**Fig 1. Comparison of axial length (A) and spherical equivalent (B) in epiretinal membrane (ERM), macular pseudohole (MPH), ERM-foveoschisis (ERM-FS) and lamellar macular hole (LMH).** The mean axial length was longer in MPH, ERM-FS, and LMH compared to ERM. The spherical equivalent was significantly smaller in the MPH, ERM-FS, and LMH compared to the ERM. Significantly different in Tukey-Kramer test, *p<0.05, **p<0.01.

in all groups, with a significant difference found between the ERM and LMH, which indicates that the postoperative BCVA was better in ERM versus that in LMH.

We compared the OCT findings among the four groups (Table 2). Inner retinal cysts were significantly more frequent in ERM-FS than in the ERM. Outer retinal cysts were significantly more frequent in the ERM-FS versus the other three groups and were more frequent in the MPH and LMH compared to the ERM. EP was significantly more frequent in the LMH compared to the other three groups, and was more frequently observed in the MPH and ERM-FS versus the ERM. EZ disruption was significantly more frequent in the LMH compared to the other three groups. Both the CFT and CRT were significantly higher in the ERM versus the other three groups (Fig 3A and B). Additionally, the ERM-FS showed significantly higher CFT and CRT compared to the LMH. The MV was significantly larger in ERM compared to the LMH (Fig 3C).

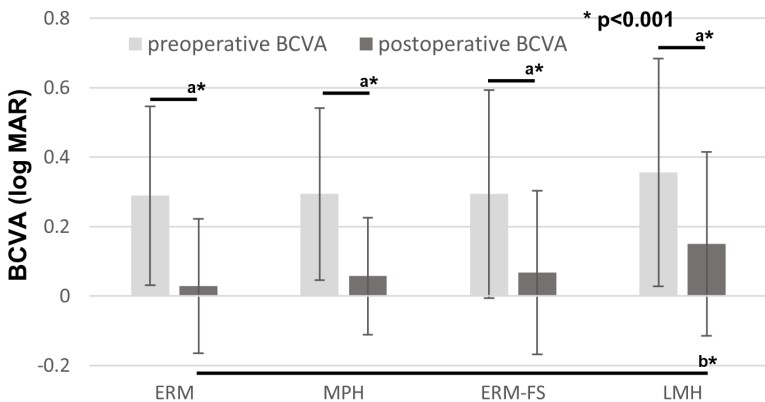

**Fig 2. Best-corrected visual acuity (BCVA) before and after surgery in epiretinal membrane (ERM), macular pseudohole (MPH), ERM-foveoschisis (ERM-FS) and lamellar macular hole (LMH). a: Significantly different in the Wilcoxon's signed-rank test. b: Significantly different in the Tukey-Kramer test.** Preoperative BCVA did not significantly differ among the groups. Postoperative BCVA was significantly worse in LMH compared to ERM.

**Table 2. OCT findings of epiretinal membrane (ERM), macular pseudohole (MPH), ERM-foveoschisis (ERM-FS) and lamellar macular hole (LMH).**

|  | ERM | MPH | ERM-FS | LMH |
|---|---|---|---|---|
| **Presence of inner retinal cyst** | 69 (11.7%) [a*] | 13 (17.1%) | 19 (27%) [a*] | 9 (21.4%) |
| **Presence of outer retinal cyst** | 80 (13.5%) | 24 (31.6%) | 57 (90.5%) [a*] | 23 (54.8%) |
| **Presence of EP** | 11 (1.9%) | 17 (22.3%) | 17 (27%) | 25 (59.5%) [a*] |
| **EZ disruption** | 67 (11.3%) | 24 (31.6%) | 25 (39.7%) | 29 (69%) [a*] |
| **CFT (µm)** | 434±97 [b,‡] | 233±94 | 261±96 [b,†] | 205±77 [b,†] |
| **CRT (µm)** | 432±84 [b,‡] | 349±77 | 387±77 [b,†] | 325±74 [b,†] |
| **MV (mm³)** | 9.8±1.5 [b*] | 9.3±1.4 | 9.6±1.4 | 9.1±1.0 [b*] |

EP = epiretinal proliferation, EZ = ellipsoid zone, CFT = central foveal thickness, CRT = central retinal thickness, MV = macular volume.

[a]: Significantly different from the other groups (*$p < 0.001$, Chi-square test). [b]: Significantly different from the other groups (*$p < 0.05$, †$p < 0.01$, ‡$p < 0.001$, Tukey-Kramer test).

Table 3 shows the relationships in each group between the pre- and postoperative BCVA and OCT findings. Presence of inner and outer retinal cyst and EZ disruption was significantly associated with both worse pre- and postoperative BCVA in the ERM. In addition, both the pre- and postoperative BCVA were significantly correlated with CFT and CRT, which demonstrated that patients with higher CFT or CRT had worse preoperative BCVA in ERM. For the MPH group, the presence of EP exhibited a significant association with worse postoperative BCVA. Furthermore, eyes with EZ disruption were associated with a worse postoperative BCVA in the LMH. No significant relationships were found between the BCVA and OCT findings in ERM-FS.

Table 4 shows the results of the multiple linear regression analysis that examined factors associated with the postoperative BCVA in each group. A worse preoperative BCVA was significantly associated with a worse postoperative BCVA in all of the groups. Presence of EZ disruption was also significantly associated with a worse postoperative BCVA in both the ERM and LMH. The presence of EP exhibited a significant association with a worse postoperative BCVA in MPH. In addition, a higher CRT was also associated with a worse postoperative BCVA in ERM-FS.

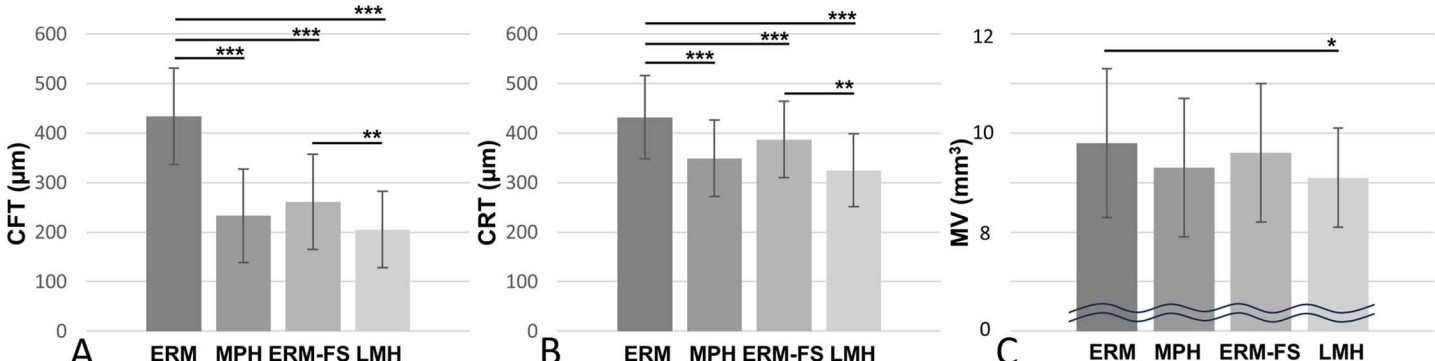

**Fig 3. Comparison of central foveal thickness (CFT) (A), central retinal thickness (CRT) (B), and macular volume (MV) (C) in epiretinal membrane (ERM), macular pseudohole (MPH), ERM-foveoschisis (ERM-FS) and lamellar macular hole (LMH) Both the CFT and CRT were significantly higher in the ERM compared to the other three groups.** Additionally, the ERM-FS showed significantly higher CFT and CRT compared to the LMH. The MV was significantly larger in ERM compared to the LMH. Significantly different from the other groups (*p < 0.05, **p < 0.01, ***p < 0.001, Tukey-Kramer test).

**Table 3. Relationships between preoperative OCT findings and visual acuity in epiretinal membrane (ERM), macular pseudohole (MPH), ERM-foveoschisis (ERM-FS) and lamellar macular hole (LMH).**

|  | ERM | | MPH | | ERM-FS | | LMH | |
|---|---|---|---|---|---|---|---|---|
|  | Pre-BCVA | Post-BCVA | Pre-BCVA | Post-BCVA | Pre-BCVA | Post-BCVA | Pre-BCVA | Post-BCVA |
| **Presence of inner retinal cyst** | 0.001‡a | < 0.001‡a | 0.407 | 0.502 | 0.337 | 0.534 | 0.432 | 0.976 |
| **Presence of outer retinal cyst** | < 0.05*a | < 0.005†a | 0.539 | 0.640 | 0.352 | 0.121 | 0.969 | 0.789 |
| **Presence of EP** | 0.317 | 0.107 | 0.288 | <0.05*a | 0.079 | 0.160 | 0.146 | 0.051 |
| **EZ disruption** | 0.001‡a | < 0.001‡a | 0.962 | 0.128 | 0.832 | 0.100 | 0.524 | < 0.05*a |
| **CFT** | < 0.05*b | < 0.05*b | 0.675 | 0.413 | 0.918 | 0.399 | 0.108 | 0.515 |
| **CRT** | < 0.05*b | < 0.01†b | 0.330 | 0.466 | 0.947 | 0.683 | 0.883 | 0.930 |
| **MV** | 0.181 | 0.113 | 0.986 | 0.218 | 0.545 | 0.296 | 0.692 | 0.888 |

BCVA = best corrected visual acuity, EP = epiretinal proliferation, EZ = ellipsoid zone, CFT = central foveal thickness, CRT = central retinal thickness, MV = macular volume.

a: Significantly correlated with pre- and/ or postoperative BCVA (*p < 0.05, † p < 0.005, ‡ p ≤ 0.001, Mann-Whitney U test).

b: Significantly correlated with pre- and/ or postoperative BCVA (*p < 0.05, † p < 0.01, Spearman's rank correlation test).

## Discussion

This study evaluated the clinical characteristics and OCT findings for ERM, MPH, ERM-FS, and LMH, and then compared these findings between the groups, along with investigating factors that affect the postoperative visual acuity. Although results showed that the association varied for the OCT findings and postoperative BCVA, there was improvement in all four groups for the postoperative BCVA. Thus, it is important to identify preoperative characteristics that may have a predictive value so that they can be used to provide the appropriate timing for the surgery.

The associated diseases, MPH, ERM-FS, and LMH, were found to be significantly more myopic than ERM. Previous studies have also reported mild to high myopia in MPH, ERM-FS, and LMH [3,5]. It has also been suggested that MPH, ERM-FS, and LMH may be associated with myopia, similarly to other myopic conditions such as myopic traction maculopathy, foveoschisis, maculoschisis, retinoschisis, retinal/foveal detachment, and full-thickness macula holes [11].

In the present study, ERM with the presence of inner and outer retinal cysts exhibited a significantly worse pre- and postoperative BCVA. Previous studies have also analyzed the relationship between visual outcomes and the

**Table 4. Multiple regression analysis between postoperative best-corrected visual acuity (BCVA) and preoperative BCVA and OCT findings in epiretinal membrane (ERM), macular pseudohole (MPH), ERM-foveoschisis (ERM-FS) and lamellar macular hole (LMH).**

| | ERM | | MPH | | ERM-FS | | LMH | |
|---|---|---|---|---|---|---|---|---|
| | Standardized coefficient | P value | Standardized coefficient | P value | Standardized coefficient | P value | Standardized coefficient | P value |
| Preoperative BCVA | 0.470 | < 0.001 | 0.488 | < 0.001 | 0.487 | < 0.001 | 0.504 | < 0.001 |
| Presence of inner retinal cyst | 0.068 | 0.210 | −0.126 | 0.263 | −0.140 | 0.248 | 0.000 | 0.998 |
| Presence of outer retinal cyst | −0.021 | 0.692 | 0.168 | 0.110 | 0.087 | 0.490 | −0.122 | 0.404 |
| Presence of EP | N.S. | – | 0.274 | < 0.05 | 0.135 | 0.308 | 0.122 | 0.383 |
| EZ disruption | 0.115 | < 0.05 | 0.190 | 0.081 | 0.204 | 0.096 | 0.342 | < 0.05 |
| CFT | N.S. | – | N.S. | – | 0.030 | 0.838 | 0.236 | 0.164 |
| CRT | 0.035 | 0.445 | 0.053 | 0.622 | 0.375 | < 0.05 | 0.109 | 0.596 |
| MV | N.S. | – | N.S. | – | −0.249 | 0.152 | 0.013 | 0.931 |

EZ = ellipsoid zone, CFT = central foveal thickness, CRT = central retinal thickness, MV = macular volume, N.S. = not selected

Values were calculated using a multiple linear regression model.

During surgery, a retinal hole was detected in 61 patients. These cases were treated with laser photocoagulation and air tamponade. None of the patients experienced postoperative complications that could lead to visual acuity loss.

presence of intraretinal cysts in ERM [12–14]. One of these studies [12] reported that intraretinal cysts did not affect postoperative BCVA at 12 months after surgery, while other studies [13,14] found the presence of cysts before surgery to be a poor visual prognostic factor, similar to our present findings. Furthermore, some of these previous studies assessed cysts following surgery [13,15,16], and suggested that microcysts in the INL were induced shortly after surgery and can have long-term effects on the visual outcomes. Outer cysts that were observed in Henle's fiber layer and the outer plexiform layer were likely due to capillary leakage caused by the mechanical traction of ERM. In contrast, inner retinal cysts were located in the INL, where the Müller cell nuclei are present, which suggests that these cysts were the result of Müller cell death. In cases of ERM with high CRT, large inner retinal cysts were observed, indicating Müller cell death, and which may be irreversible through surgery. Thus, this may explain why ERM with inner and outer cysts had worse pre- and postoperative BCVA. Further analysis of the relationship between cysts and BCVA could provide more reliable insights by evaluating both the cysts that remain after surgery and those induced by the surgical procedure.

The CFT and CRT were significantly higher in ERM. Furthermore, eyes with increased CFT and CRT had worse pre- and postoperative BCVA. While some studies reported correlations between CFT or CRT and poor BCVA, others have not found such correlations [12,17–19]. More recently, it has been proposed that the thickness of the ectopic inner foveal layers is a factor related to the visual acuity [19–21]. In future research, it will be necessary to examine which specific retinal layers contribute to the increased CFT and CRT. Therefore, we suggest that the presence of the inner and outer cyst, as well as the thickness of the CFT and CRT, should be evaluated when determining the timing of surgery in ERM.

The results of the present study also showed that the presence of inner retinal cysts was significantly more frequent in ERM-FS than in ERM. A previous study has also reported that microcystoid spaces in the INL were often present in ERM-FS, with the Müller cell dysfunction suggested to play a role in the development of the microcystoid spaces [22]. The presence of outer retinal cysts was significantly more frequent in ERM-FS than in the other three groups. Hetzel et al. [4] reported finding outer retinal cysts in 52.9% of eyes with ERM-FS, and noted that the BCVA was significantly lower in eyes with intraretinal cystoid spaces. In the present study, we found that the presence of intraretinal cysts before surgery was not correlated with the preoperative or postoperative BCVA.

The postoperative BCVA was significantly better in the ERM group compared to the LMH group. However, preoperative BCVA showed that there was no significant difference among the four groups. Previous studies have also reported finding no significant differences in the preoperative BCVA between ERM-FS and LMH [23], as well as among LMH, ERM-FS, and MPH [3]. As preoperative BCVA is a crucial factor in determining the need for surgery, it may not differ significantly between these diseases. For the postoperative BCVA, Pertile et al. [23] found that the BCVA improvement occurred significantly earlier in eyes with ERM-FS compared to LMH. Similarly, Omoto et al. [5] reported that while postoperative BCVA did not improve at 3 months postoperatively, it did improve at the final visit for LMH. Mohammed et al. [24] compared BCVA outcomes in MPH, ERM-FS, and LMH, and reported that at the final visit, BCVA improved in MPH and ERM-FS but not in LMH after surgery. Although the BCVA could potentially improve during a longer follow-up period, LMH may represent a stage beyond the other disorders, in which irreversible morphological changes could have occurred.

The presence of EP was significantly more frequent in LMH. Previous studies have also reported finding a high prevalence of EP, ranging from 36% to 100% [3,25–27]. Some of these studies also reported on the predicted origin of EP. Yang et al. [27] suggested that EP originated from Müller cells, which was based on immunohistochemical analyses. Our observations support this suggestion, as the EP appears yellowish during surgery, which is likely due to the xanthophyll pigments that are stored in the Müller cells. Since LMH shows tissue loss at the fovea, EP may develop as a compensatory response of gliosis of the Müller cells.

The present results showed that MPH patients with EP had significantly worse postoperative BCVA compared to those without EP. In our study, we peeled the ILM towards the edges of the fovea and then inverted it over the fovea after ERM peeling, if EP was present. In MPH, EP is located above the ERM, which makes it difficult for the EP to directly contact the retina. In contrast, EP often exists above the ILM in LMH, as most LMH cases lack ERM. Thus, this provides more opportunities to directly attach to the retina by embedding EP and ILM. If EP develops as a compensatory response to tissue loss, it should be removed from the ERM or ILM and attached to the retina by embedding EP. This may explain why the postoperative BCVA in MPH with EP was worse compared to that without EP, while no significant difference was observed in LMH between with and without EP.

EZ disruption was significantly more frequent in the LMH group compared to the ERM group. In addition, the presence of EZ disruption in the LMH was associated with worse postoperative BCVA in the present study. Previous studies have analyzed the predictive value of EZ disruption in ERM, and have reported that patients with preoperative EZ disruption had significantly worse postoperative BCVA [12,18]. In contrast, some studies reported finding no significant effect of the preoperative EZ disruption on postoperative BCVA in ERM [21,28,29]. One of these studies reported that the presence of a continuous postoperative EZ line was significantly associated with a better postoperative BCVA [21]. While EZ disruption can be restored after surgery, the process can take a considerable amount of time. As we did not assess postoperative EZ conditions in the present study, further analysis will need to be conducted to examine morphological changes of EZ after surgery.

We also evaluated the factors influencing postoperative BCVA using multivariate regression analysis. In all of the groups, postoperative BCVA was significantly associated with preoperative BCVA, which was consistent with previous reports [12,30]. In both ERM and LMH, the EZ disruption was also associated with the postoperative BCVA, which indicates that the EZ condition should also be considered when predicting the visual outcomes after surgery. Not only preoperative BCVA but also the presence of EP was correlated with postoperative BCVA in MPH in our results.

In conclusion, this study demonstrated that vitrectomy improved visual outcomes in ERM, MPH, ERM-FS, and LMH, although the OCT findings were different for each disease. Therefore, preoperative BCVA and OCT findings need to be carefully evaluated to comprehensively predict the postoperative BCVA. Furthermore, to provide a more detailed prediction of the postoperative visual acuity, postoperative OCT findings will also need to be carefully evaluated. At the present time, although surgery for these four diseases is fundamentally vitreoretinal surgery, specific techniques (such as whether

to completely peel the ILM, invert it, or preserve it without peeling, as well as whether to remove or embed the EP when present) need to be determined based on the OCT findings to achieve better surgical outcomes.

## Supporting information

**S1 Data. Raw data for all parameters.**
(XLSX)

## Author contributions

**Conceptualization:** Fumiki Okamoto.

**Data curation:** Noriko Kubota, Yuji Nakano, Hitoshi Goto.

**Formal analysis:** Noriko Kubota.

**Investigation:** Kazunori Miyata, Yosai Mori.

**Project administration:** Fumiki Okamoto.

**Resources:** Kazunori Miyata, Yosai Mori.

**Supervision:** Fumiki Okamoto.

**Writing – original draft:** Noriko Kubota.

**Writing – review & editing:** Noriko Kubota, Fumiki Okamoto.

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
