## [Decision Letter · Decision Letter 0]

8 Apr 2025

PONE-D-25-12447Clinical Characteristics and Optical Coherence Tomography Findings in Epiretinal Membrane, Macular Pseudohole, Epiretinal Membrane-foveoschisis, and Lamellar Macular HolePLOS ONE

Dear Dr. Kubota,

Thank you for submitting your manuscript to PLOS ONE. After careful consideration, we feel that it has merit but does not fully meet PLOS ONE’s publication criteria as it currently stands. Therefore, we invite you to submit a revised version of the manuscript that addresses the points raised during the review process.

We look forward to receiving your revised manuscript.

Kind regards,

Jiro Kogo

Academic Editor

PLOS ONE

Journal Requirements:

Reviewers' comments:

Reviewer's Responses to Questions

**Comments to the Author**

1. Is the manuscript technically sound, and do the data support the conclusions?

Reviewer #1: Yes

Reviewer #2: Yes

2. Has the statistical analysis been performed appropriately and rigorously? 

Reviewer #1: Yes

Reviewer #2: Yes

3. Have the authors made all data underlying the findings in their manuscript fully available?

Reviewer #1: Yes

Reviewer #2: No

4. Is the manuscript presented in an intelligible fashion and written in standard English?

Reviewer #1: Yes

Reviewer #2: Yes

5. Review Comments to the Author

Reviewer #1: This study investigates the clinical characteristics, postoperative prognosis, and related factors of ERM, MPH, ERM-FS, and LMH in a relatively large cohort. The findings are highly valuable and contribute significantly to the field.

I have a few questions and suggestions for clarification:

1. How was macular volume measured and calculated? Please provide details on the methodology used.

2. The manuscript states that the ILM was inverted in cases of LMH with EP. How was the EP itself managed? Was it spared, embedded, or removed?

Additionally, the statement in line 268, "In our study, we peeled the ILM~," should be included in the Methods section for clarity and consistency.

3.It would be beneficial to include data on the extent of EZ disruption improvement in each postoperative group. If such an analysis has not yet been conducted, I encourage the authors to consider it in future studies.

Overall, this study provides important insights, and addressing these points would further enhance the clarity and impact of the findings.

Reviewer #2: This paper by Kubota et al. provides a detailed examination of ERM and related diseases through analysis of a large number of cases.

The paper would be improved by making the following revisions.

Major comments

L82-87 How did the authors classify cases where it was difficult to determine the type of disease, i.e. cases that appeared to be LMH or FS depending on the scan?

L87-90 Did the authors exclude cases in which a retinal hole was seen during surgery, even though it was not apparent before surgery?

L99-103 Are the parameters such as CST, CRT, and MV measured automatically or manually? Also, how many micrometers from the fovea is the area being measured?

Also, if they are measured manually, how many researchers measured them?

L127-128 Was there a significant difference in the rate of cataract surgery for each disease between the groups? The rate of cataract surgery may be related to the improvement in postoperative visual acuity.

In Table 1, axial length and spherical equivalent, and in Table 2, CRT, CFT, and MV, there is a significant difference according to the Tukey Krumer test, but it is difficult to see from the tables which groups have a significant difference. I advise you to create a graph to make it easier to understand visually.

Minor comments

Is the extent of ILM peeling standardized for the cases in this study?

If intraocular tamponade material was used during surgery, please add a breakdown of this to Table 1.

Is OCT evaluated from multiple directions, both horizontally and vertically?

Figure 1 does not have a legend, so this needs to be added.

6. PLOS authors have the option to publish the peer review history of their article (what does this mean? ). If published, this will include your full peer review and any attached files.

**Do you want your identity to be public for this peer review?** For information about this choice, including consent withdrawal, please see our Privacy Policy .

Reviewer #1: No

Reviewer #2: No

---

## [Author Response · Author response to Decision Letter 0]

15 Apr 2025

April 15, 2025

Jiro Kogo

Academic Editor

PLOS ONE

Dear Dr. Kogo and reviewers,

Thank you very much for the opportunity to revise and submit our manuscript entitled, “Clinical Characteristics and Optical Coherence Tomography Findings in Epiretinal Membrane, Macular Pseudohole, Epiretinal Membrane-foveoschisis, and Lamellar Macular Hole” to PLOS ONE. We are sincerely grateful for the time and effort you and each of the reviewers have invested in evaluating our work and providing thoughtful, constructive feedback. We have carefully revised the manuscript in accordance with your valuable comments and suggestions. It is with great appreciation that we resubmit our revised manuscript for your further consideration. We believe the changes we have made have improved the quality and clarity of the paper, and we hope that our responses and revisions satisfactorily address all concerns raised.

To facilitate your review, we have included a detailed, point-by-point response to the questions and comments delivered in your letter dated April 9, 2025.

Reviewer #1

1. How was macular volume measured and calculated? Please provide details on the methodology used.

Thank you for pointing this out. We have added details regarding the measurement of macular volume in the Method section. Specifically, macular volume (MV) was measured using the automated function of the spectral-domain OCT within a 6 mm diameter centered on the fovea.

2. The manuscript states that the ILM was inverted in cases of LMH with EP. How was the EP itself managed? Was it spared, embedded, or removed?

Additionally, the statement in line 268, "In our study, we peeled the ILM~," should be included in the Methods section for clarity and consistency.

We clarified that EP was embedded on the fovea. This information has now been added to both the Methods and Discussion sections. The procedural statement from line 268 has also been included in the Methods section.

3. It would be beneficial to include data on the extent of EZ disruption improvement in each postoperative group. If such an analysis has not yet been conducted, I encourage the authors to consider it in future studies.

Thank you for your insightful suggestion. While we were unable to include postoperative findings related to EZ disruption improvement in the present study, we agree with the importance of this analysis. We plan to investigate this aspect in future studies.

Reviewer #2

Major comments

1. L82-87 How did the authors classify cases where it was difficult to determine the type of disease, i.e. cases that appeared to be LMH or FS depending on the scan?

We used both horizontal and vertical B-scan OCT images to diagnose ERM, MPH, ERM-FS, and LMH. Eyes that satisfied the diagnostic criteria for more than two associated diseases were classified as overlapping cases. Those overlapping cases were included in the analysis of all corresponding disease groups.

2. L87-90 Did the authors exclude cases in which a retinal hole was seen during surgery, even though it was not apparent before surgery?

Yes, we did. A total of 61 eyes were found to have a macular hole during surgery. These cases were treated with laser photocoagulation and air tamponade. None of these patients experienced postoperative complications that would result in visual acuity loss. This information has been added to the Results section.

3. L99-103 Are the parameters such as CST, CRT, and MV measured automatically or manually? Also, how many micrometers from the fovea is the area being measured?

Also, if they are measured manually, how many researchers measured them?

Thank you for this important question. We clarified in the Methods section that these measurements were conducted using the automated function of the spectral-domain OCT.

4. L127-128 Was there a significant difference in the rate of cataract surgery for each disease between the groups? The rate of cataract surgery may be related to the improvement in postoperative visual acuity.

We analyzed the rate of cataract surgery among the disease groups and found no significant difference (p = 0.41). This result has been added to the Results section.

5. In Table 1, axial length and spherical equivalent, and in Table 2, CRT, CFT, and MV, there is a significant difference according to the Tukey Krumer test, but it is difficult to see from the tables which groups have a significant difference. I advise you to create a graph to make it easier to understand visually.

Thank you for this valuable suggestion. We have added graphs illustrating the differences in axial length, spherical equivalent, CRT, CFT, and MV for better visual interpretation of the data.

Minor comments

1. Is the extent of ILM peeling standardized for the cases in this study?

Yes, it is. In all cases, the ILM was peeled in an area approximately equal to or double the vertical extent of the optic disc. This detail has been added to the Methods section.

2. If intraocular tamponade material was used during surgery, please add a breakdown of this to Table 1.

Thank you for this suggestion. A breakdown of the intraocular tamponade materials used has been added to Table 1.

3. Is OCT evaluated from multiple directions, both horizontally and vertically?

Yes, it is. Horizontal and vertical OCT B-scan images were evaluated for each case. This has been added to the Methods section.

4. Figure 1 does not have a legend, so this needs to be added.

Thank you for pointing this out. We have added a figure legend for Figure 1.

Finally, we have made a minor correction to Reference 9 in the reference list.

Once again, we sincerely thank you for giving us the opportunity to revise and strengthen our manuscript with your valuable comments and queries. We deeply appreciate the valuable comments and insightful suggestions provided by the reviewers. We have made every effort to incorporate the feedback and improve the clarity, quality, and scientific rigor of our work. We hope that the revisions presented will meet your expectations and support the acceptance of our manuscript for publication.

Sincerely yours,

Noriko Kubota, MD

Department of Ophthalmology, Nippon Medical School

1-1-5 Sendagi, Bunkyo-ku, Tokyo 113-8603, Japan.

Phone: +81-3-3822-2131, Fax: +81-3-5685-0988, E-mail: oishinoriko@nms.ac.jp

---

## [Editor Report · Decision Letter 1]

17 Apr 2025

Clinical Characteristics and Optical Coherence Tomography Findings in Epiretinal Membrane, Macular Pseudohole, Epiretinal Membrane-foveoschisis, and Lamellar Macular Hole

PONE-D-25-12447R1

Dear Dr..Kubota

We’re pleased to inform you that your manuscript has been judged scientifically suitable for publication and will be formally accepted for publication once it meets all outstanding technical requirements.

Kind regards,

Jiro Kogo

Academic Editor

PLOS ONE
---

## [Editor Report · Acceptance letter]

PONE-D-25-12447R1

PLOS ONE

Dear Dr. Kubota,

I'm pleased to inform you that your manuscript has been deemed suitable for publication in PLOS ONE. Congratulations! Your manuscript is now being handed over to our production team.

Kind regards,

on behalf of

Prof. Jiro Kogo

Academic Editor

PLOS ONE